# Quantitative Analytical and Computational Workflow for Large-Scale Targeted Plasma Metabolomics

**DOI:** 10.3390/metabo13070844

**Published:** 2023-07-13

**Authors:** Antonia Fecke, Nay Min Min Thaw Saw, Dipali Kale, Siva Swapna Kasarla, Albert Sickmann, Prasad Phapale

**Affiliations:** 1Leibniz-Institut für Analytische Wissenschaften—ISAS—e.V., Otto-Hahn-Str. 6b, 44227 Dortmund, Germany; antonia.fecke@isas.de (A.F.); naymin.saw@isas.de (N.M.M.T.S.); dipali.kale@isas.de (D.K.); siva.kasarla@isas.de (S.S.K.); albert.sickmann@isas.de (A.S.); 2Department Hamm 2, Hochschule Hamm-Lippstadt, Marker-Allee 76-78, 59063 Hamm, Germany

**Keywords:** metabolomics, metabolite quantification, LC-MS, quantitative spectral library, relative response factor

## Abstract

Quantifying metabolites from various biological samples is necessary for the clinical and biomedical translation of metabolomics research. One of the ongoing challenges in biomedical metabolomics studies is the large-scale quantification of targeted metabolites, mainly due to the complexity of biological sample matrices. Furthermore, in LC-MS analysis, the response of compounds is influenced by their physicochemical properties, chromatographic conditions, eluent composition, sample preparation, type of MS ionization source, and analyzer used. To facilitate large-scale metabolite quantification, we evaluated the relative response factor (RRF) approach combined with an integrated analytical and computational workflow. This approach considers a compound’s individual response in LC-MS analysis relative to that of a non-endogenous reference compound to correct matrix effects. We created a quantitative LC-MS library using the Skyline/Panorama web platform for data processing and public sharing of data. In this study, we developed and validated a metabolomics method for over 280 standard metabolites and quantified over 90 metabolites. The RRF quantification was validated and compared with conventional external calibration approaches as well as literature reports. The Skyline software environment was adapted for processing such metabolomics data, and the results are shared as a “quantitative chromatogram library” with the Panorama web application. This new workflow was found to be suitable for large-scale quantification of metabolites in human plasma samples. In conclusion, we report a novel quantitative chromatogram library with a targeted data analysis workflow for biomedical metabolomic applications.

## 1. Introduction

Metabolomics research plays a crucial role in understanding complex biochemical pathways and identifying biomarkers for various clinical and biomedical applications [1]. Untargeted or global metabolomics is a hypothesis-generating approach, which involves the profiling of metabolites without prior knowledge and has provided important insights into several disease and drug response mechanisms [2,3,4]. However, metabolite quantification using an untargeted metabolomics approach often faces challenges in terms of data quality, reliability, and reproducibility of measured metabolite profiles [5,6,7]. These quantitative challenges arise from various causes, including variation in electrospray ionization (ESI) response, matrix effect, dynamic concentration range, sensitivity, and even peak curation or integration errors by software [5]. On the other hand, targeted metabolomics methods and data acquisition approaches can provide the absolute or relative quantification and validation of a chosen subset of metabolites, based on a hypothesis-driven approach [8]. Targeted metabolomics allows researchers to circumvent the shortcomings of the global approach and provides more accurate and precise results that can complement untargeted datasets [9]. Still, major limitations to performing targeted analysis are the availability of authentic standards, a blank or surrogate sample matrix, stable isotope-labeled internal standards, method development time, as well as the time required for calibration curve preparation and analysis. These limitations severely hinder the translation of LC-MS-based metabolomics methods into clinical and biomedical applications [10].

The semi-targeted or hybrid approach is increasingly being performed on high-resolution mass spectrometers (HRMS) such as orbitraps instead of conventional triple quadrupole instrumentation. This approach allows researchers to acquire data in an untargeted manner while maintaining quality controls for a targeted set of metabolites [11]. Although such a hybrid approach offers flexibility in data acquisition with less method development requirements compared to the triple quadrupole approach, it does not provide absolute quantification. In the most common external calibration (EC) methods, standards concentrations are prepared in the same matrix as the samples using an authentic reference standard of the target metabolites. Internal standards (IS) also usually need to be spiked into all the external calibration standards for normalization as well as quality control (QC) to track experimental and instrumental variability. Unlike drug analysis [12], endogenous metabolite quantification is particularly challenging due to the lack of an ideal surrogate or blank matrix free of endogenous metabolites and the availability of large-scale, stable, isotope-labeled internal standards, which makes the preparation of standard calibration curves often unattainable [13,14]. Hence, semi-targeted methods use calibration curves with serial dilutions of the analyte in neat organic or aqueous solvents, which is a routinely practiced method for relative quantitation [15]. However, such methods may not be suitable for the absolute quantification of endogenous metabolites in several biological samples with complex matrices such as biofluids, tissues, and cells. Thus, the clinical utility of semi-targeted HRMS-based metabolomics methods is still limited and not yet broadly adopted by the biomedical community.

To overcome these challenges, internal calibration (IC) has gained interest as an alternative to the classical EC approach, especially for the quantification of endogenous analytes including peptides [16], food toxins [17], water pollutants [18], and endogenous steroids [19,20]. IC eliminates the need and time to prepare an appropriate blank matrix to construct the EC curves. It relies on weighted response factor calculations to estimate the absolute quantity of endogenous metabolites without EC [19]. This IC methodology, also called the “single-point calibration”, has been applied in targeted metabolomics studies [21,22]. The metabolome coverage from these studies is limited to a few metabolites from a chemical class, such as four amino acids [21] and six steroids [20]. Another approach, using the standard addition method, was used for endogenous phytochemical quantification [23] and urinary metabolites [24] and has gained interest in targeted metabolomics. However, we observed it has severe limitations in terms of metabolome coverage and known issues with subtraction of endogenous metabolite levels [23].

A major challenge in the adoption of IC methods for large-scale quantification is obtaining weighted response factors or relative response factors (RRF) for hundreds of endogenous metabolites. For example, over 200 metabolites can be routinely quantified from clinically important biofluids, such as blood plasma and urine metabolomics [15]. Since these methods do not consider the matrix effect, as ECs are prepared in buffers or solvents, IC can provide a powerful approach for absolute metabolite quantification. As RRFs are influenced by the ionization efficiencies of individual compounds, a computational approach has been used for their large-scale predictions [25,26,27]. The RRF serves as a correction factor for matrix effects relative to a non-endogenous reference compound and considers the compound’s specific response in LC-MS analysis. Therefore, it can combine advantages of EC and stable isotope-based quantification. However, a challenge still remains regarding their transferability among different mass spectrometers [25], chromatographic systems, mobile phase pH, and other experimental factors [27]. Thus, the overall targeted quantification of metabolites from biological samples remains a challenge, which is a major limiting factor for its adoption in clinical practices.

In this study, we addressed the above challenges and developed a novel analytical and computational workflow for the large-scale quantification of metabolites in human plasma. First, we compared and optimized a new zwitterionic sulfobetaine stationary phase (Z-HILIC)-based [28,29] LC-MS method for the detection (and quantification) of over 300 (and 100) clinically relevant metabolites. This stationary phase was found to be superior compared to the widely used amide HILIC method in terms of chromatography of polar metabolites. Next, we evaluated the RRF-based quantitative approach using multiple reference compounds and compared it with conventional EC quantification in human plasma samples. The quantification results were benchmarked with reports for human plasma standard reference material. To facilitate automated data analysis, metabolite quantification, and sharing of results, we utilized the Skyline–Panorama workflow [30]. Finally, the results are compiled as the first-ever “Quantitative Metabolomics Library (IQML)”, which has been made publicly available. The overall analytical and computational workflow, along with the published library, can provide a platform for metabolomics research to adopt this novel quantitative metabolomics approach for several other biological sample types. Although this approach using predetermined RRFs shows great potential in many applications, such as biomarker quantification, liquid biopsies, and clinical diagnosis, especially in cases where authentic matrices and standards are difficult to obtain, we have also highlighted its limitations, scope, and outlook.

## 2. Materials and Methods

### 2.1. Materials, Chemical, Reagent, and Standards

LC-MS-grade solvents acetonitrile (Cat. No. 0001207802BS), methanol (Cat. No. 0013684102BS), and isopropanol (Cat. No. 0016260602BS) were purchased from Biosolve BV (Valkenswaard, The Netherlands); MS grade water (Cat. No. 047146.K7) was obtained from Thermo Fisher, Kandel, Germany; and dimethyl sulfoxide (Cas. No. 67-68-5) was obtained from Merck, Darmstadt, Germany. Ammonium acetate (PCode: 102326574, Merck, DE) and ammonium hydroxide (PCode: 101344936, Fluka Analytical, Schwerte, Germany) were used as buffers and reagents. Metabolite standards were purchased from Metasci, Toronto, Canada. Additionally, the glycolysis/gluconeogenesis metabolite library (PCode: 1003025785) was obtained from Merck, Darmstadt, Germany (Table 1). As internal standards and reference compounds for the RRF approach, the stable-isotope-labeled amino acid mixture for mass spec (Product No. 909653, Table 1) and as biological sample pooled human plasma (Product No. P9523) was purchased from Merck, Darmstadt, Germany. Metabolite standards stock solutions for external calibrations were prepared at a concentration of 1 mg/mL (Appendix A) and used for further dilutions. The stocks were stored at −80 °C. Two chromatographic columns used for method development were purchased from Waters Corp (Milford, MA, USA). A comparison was made between the XBridge BEH Amide XP (2.5 µm, 130 Å, 100 × 2.1 mm) and the Atlantis Premier BEH Z-HILIC (5 µm, 89 Å, 100 × 2.1 mm) columns.

### 2.2. Preparation of Standards

The 281 metabolite standards were prepared by weighing 0.6 to 2 mg into 2 mL Eppendorf tubes followed by their dilution to prepare a stock solution of 1 mg/mL (Appendix A). The metabolites were then divided into 33 sets, each containing up to 10 metabolite standards. An amount of 100 µL of each stock solution were pipetted into a new 2 mL Eppendorf tube and filled with acetonitrile:water (90:10), if necessary, to achieve a concentration of 100 µg/mL of each standard in each set. These set solutions were then transferred into a 96-well plate and diluted with acetonitrile:water (90:10) into concentrations of 100 µg/mL, 50 µg/mL, 25 µg/mL, 10 µg/mL, and 1 µg/mL. The 96-well plates were then stored in a −20 °C freezer until further analysis. These solutions were used for LC-MS method development on the Xbridge Amide column. For z-HILIC column analysis, 100 metabolite standard solutions were prepared in 5 sets (20 compounds each) of equimolar concentrations from the previous 1 mg/mL stock solution. Additionally, a labeled amino acid mix was reconstituted in 1 mL ultrapure water as per vendor instructions. The amount of volume needed to prepare a 5 mL of 200 µM solution was calculated and pipetted for each metabolite. Into each well of a 96-well plate, 10 µL of the internal standard was added. Afterwards, the 200 µM stock mix was added and diluted to eight different concentrations (0.03 µM, 0.3 µM, 3 µM, 6 µM, 12 µM, 25 µM, 50 µM, 100 µM). The final concentration of stable-isotope-labeled internal standards was 1.25 µM in each well. A system conditioning sample was created by mixing all five standard mixes into one well. The standards have been measured in triplicates.

### 2.3. Methods

#### 2.3.1. Plasma Sample Preparation

For the preparation of pooled plasma samples commercially available, lyophilized powder was reconstituted in water as per vendor instructions. An aliquot of 50 µL reconstituted plasma was transferred into an Eppendorf tube. Internal standards (stable-isotope-labeled amino acids) were added to the final concentration of 1.25 µM, and 300 µL of acetonitrile: methanol (95:5) was added as the extraction solvent. After vortexing in Thermomixer comfort (Eppendorf, Hamburg, Germany) for 10 min at ~4 °C, the samples were stored in a −80 °C deep freezer (TwinGuard, Panasonic, Kadoma, Japan) overnight for incubation to ensure complete protein precipitation. Afterward, the samples were centrifuged at 20,000 rcf for 20 min at ~4 °C (Centrifuge 5424 R, Eppendorf, Hamburg, Germany). The supernatants were collected and dried under the in-house assembled nitrogen evaporator. For reconstitution, 100 µL of acetonitrile: water (50:50) was added. The samples were again vortexed for 10 min at 4 °C and stored at −80 °C for one hour of incubation to check any further precipitation or particulate matters. Lastly, after centrifugation at the same conditions as before, the supernatant was collected for LC-MS analysis. Three technical replicates were prepared for method validation. Respective diluents from sample preparations were used as blank samples to check background peaks and instrument noise.

#### 2.3.2. LC-MS Method

LC-MS measurements were performed on the Vanquish Duo UHPLC-system (ThermoFisher, Waltham, MA, USA) equipped with a dual pump, autosampler, and a thermostatic column compartment with the XBridge BEH Amide XP Column or Atlantis Premier BEH Amide Z-HILIC column. The mobile phases used were 100% water + 0.1% ammonium acetate + 0.1% ammonium hydroxide (A) and 100% acetonitrile (B) at a flow rate of 0.4 mL/min. Initially, the mobile phase composition was 95% solvent B. After 0.5 min, it was linearly decreased to 80% (0.5 to 1.5 min) and held (1.5 min to 7.7 min). Afterward, the percentage B was further decreased to 70% (7.7 min to 9.5 min) and from there to 10% (9.5 min to 10.5 min). The percentage was held for 1.5 min (10.5 min to 12 min) and increased again to 30% (12–16 min). From here, the percentage of B increases to the original composition of 95% (16 min to 16.5 min) and was held until the end of the run (19 min). The mass spectrometer used for this study was Orbitrap Exploris 240 (ThermoFisher, Waltham, MA, USA) with an ESI spray source. Measurements were performed in dual ion mode at a resolution of 120,000 and a HCD collision energy of 35% with a mass range of 65–850 m/z. The DDA scan number was set to fragment the 5 most intense masses. The HSI source was operated at a voltage of 3500 V (HESI+) and 3000 V (HESI-). Sheath gas was set to 35 units, aux gas to 10 units, and sweep gas was not used. The ion transfer tube was heated to 300 °C; the vaporizer temperature was set to 310 °C.

#### 2.3.3. AutoQC

To monitor the system suitability prior to the experiment, an AutoQC method was introduced using Panorama web. For this, the amino acid standard is diluted 1:10 using acetonitrile: water (90:10) and measured regularly at the same concentration. The data are automatically uploaded to Panorama web. Here, it is compared to previous measurements to see any changes in parameters such as retention time, mass accuracy, TIC, and peak width.

### 2.4. Data Analysis

#### 2.4.1. Peak Extraction and Processing with Skyline

Skyline (version 22.2) is an open-source software well known in proteomics [31] for its unique features, especially visualization and advanced method refinement options, such as retention time scheduling and reproducibility. These features are now also implemented for small molecule workflows such as pharmaceuticals (drug metabolism and toxicology), metabolomics, forensics, and food safety [32]. Full-scan (MS1) features were set to Orbitrap for the precursor mass analyzer, resolving power of 120,000 at 400 m/z, and one isotopic peak. The instrument scan range was set to 65–850 m/z. Raw data files were imported directly into Skyline, and ion intensity chromatograms were displayed for a single isotopic peak. The Skyline chromatography template file can be viewed and downloaded via the Panorama Public data repository [30].

#### 2.4.2. Statistical Analysis

The calibration curves were calculated with linear regression with the weighting 1/(x) function from Skyline. The R^2^ (coefficient of regression), CV (coefficient of variation) distribution, and quantification of metabolites were represented as a histogram or box plot using ggplot in R program [33].

#### 2.4.3. Method Validation Parameter Calculations

To evaluate the linearity, dynamic range and accuracy, the standard calibration curves of metabolites with 8 concentration ranges between 0.03 to 100 µM were prepared in triplicate as described in Methods. Skyline software was used for peak integration and to plot and estimate linearity, equation of line, and regression coefficient (R^2^) using this 8-point calibration curve for each metabolite. The peak areas for replicates were used to calculate % CV for each detected metabolite. The LOD was evaluated in Skyline through extrapolation of the equation lines and is defined at the concentration in which the quantifier transition represented a signal to noise ratio (S/N) of >2. Microsoft Excel was used for Skyline data output (peak areas and annotations) to define the LOQ as the lowest concentration at which the metabolite was identified.

#### 2.4.4. Quantification Using Relative Response Factors (RRF)

Microsoft Excel was used on Skyline data output (peak areas and annotations) to define the response factor (RF). The response factor describes a compound’s specific response in LC-MS analysis. Different compounds can exhibit a different level of response at the same concentration due to their ionization efficiency. This is described by the RF. Using the slopes of the weighted line equations for each metabolite, a constant response factor was determined and used to calculate the relative response factor.

Reference compounds are required for RRF calculations. As described in the sample preparations, heavy-labeled amino acids were added as internal standards during the sample preparations. The peak area was averaged across all replicates for each ionization mode; six internal standards with the smallest coefficient of variance in their peak areas were selected as reference compounds (Appendix A). For these, an RF was determined using the ratio of the peak area and concentration. Next, the relative response factor was calculated for each metabolite by dividing the regression slope with the response factor of the reference compounds (Equation (1)).
RRF = (slope (compound))/(RF (reference compound))(1)

To evaluate the accuracy of the reference compound selection, the RRF was calculated relative to all six reference compounds for each metabolite. In total, 6 RRFs were calculated for each metabolite in both ion modes.

The RRF-based quantification has been performed using Equation (2):c(compound) = (Peak area (compound) × c(reference standard))/(Peak area (reference standards) × RRF)(2)

#### 2.4.5. Targeted Identification and Fragmentation (MS/MS) Data

All data with standards and plasma samples were collected in DDA mode with MS-MS data acquired for the top 5 most intense ions. We used Progenesis QI for identification of metabolites using our in-house spectral library [34] and the ISAS Quant metabolomics library (IQML). Although Progenesis is designed for untargeted data analysis, we customized IQML to perform targeted metabolite analysis. The Progenesis data obtained from plasma samples were also used to cross-validate quantification and peak integration performance of Skyline software.

#### 2.4.6. Data Availability

All datasets were processed with open-source software Skyline and are available publicly on Panorama Web portal as a Skyline document including extracted ion chromatograms, MS spectra, and respective summary charts. The libraries and data files can be freely downloaded from ‘ISAS Dortmund–Spatial Metabolomics’ folder and named as “ISAS_Quant_Metabolomics_library”. The MS/MS library was submitted as tags query “Metabolites identified from humnan plasma with MS/MS matched standards’’ and can be freely downloaded from MoNA—MassBank of North America website [35,36] under the name “Human_Plasma_Quant”.

## 3. Results and Discussion

The overall analytical and computational workflow, including LC-MS method development, AutoQC, Skyline, and RRF quantification, and sharing data on the Panorama web portal, is summarized in Figure 1. The results from each step of the workflow and their discussion are explained in the following sections.

### 3.1. LC-MS Method Development

First, we optimized the widely used HILIC metabolomics method with a basic mobile phase (pH) and an XBridge Premier BEH-Amide column [37]. The HILIC method was chosen to ensure adequate retention for polar, hydrophilic, or ionic compounds that cannot be properly separated using a reversed-phase (RP) LC method, as used in multiple metabolomics studies. HILIC methods combine a stationary phase similar to that of a normal-phase (NP) LC to help the retention of polar compounds while maintaining a mobile phase more similar to reversed-phase LC methods. This helps the analysis of compounds that cannot be separated using RP-LC but are not soluble in NP-LC mobile phases. We analyzed 281 metabolite standards of multiple pathways and classes (Figure 2) in dual ion mode (positive and negative ionization) at six mass concentrations ranging from 1 to 100 µg/mL. However, we observed that multiple peaks were not properly resolved and showed non-Gaussian peak shapes. To address this, we manually curated the data using Xcalibur software and categorized the detected peaks as “not detected”, “split”, “broad”, or “good” (Appendix A). The distribution of peak shapes in both ion modes and each concentration is shown in Figure 3a. The results suggest that more metabolites were detected in negative ESI, with a maximum of 60% of peaks categorized as good, while in positive ESI, only 44% of peaks were classified as good at 25 µg/mL concentration. Additionally, the evaluation showed that the number of split peaks gradually increases with concentration, indicating column overload. In negative ESI at 1 µg/mL concentration, 10% of the peaks are classified as split, while at 100 µg/mL, the percentage increases to 26% (Figure 3a). This indicates that the chosen concentration range could have an impact on the peak shape of the metabolites, as well as the varying molar masses of the compounds leading to different molar concentrations.

To address this issue (Figure 3 and Appendix A), we repeated the measurement using equimolar concentrations of previously selected metabolites (Appendix A). To ensure reliable quantification and peak integration, metabolites were selected based on three criteria: (1) it is detected in at least three concentrations classified as “good” or “broad” (Appendix A), (2) showing a linear relationship between concentration and peak area with an R^2^ > 0.9 in at least three data points, and (3) not eluting in void volume with RT > 1 min. In total, we selected 100 metabolites using these criteria and measured them at eight concentrations ranging from 0.03 to 100 µM, using the BEH Z-HILIC column. We chose this column due to its reported ability to provide sharper peak shapes while providing a similar retention mechanism [28,29].

We observed a remarkable improvement in the peak shape of polar metabolites, particularly at lower concentrations, with peaks being sharper and less noisy compared to the BEH-Amide column. An example of this can be seen in Figure 3b, which compares two examples of metabolites measured on the BEH-Amide and Z-HILIC columns. The figure shows pyridoxine and taurocholic acid in positive ESI at comparable concentrations. Both metabolites show an improved, more Gaussian, and sharper peak using the Z-HILIC column. According to Sonnenberg et al., the Z-HILIC column has a weaker electrostatic interaction due to the column chemistry relying on a 1:1 pH-independent sulfobetaine zwitterion [28]. This weaker electrostatic interaction minimizes competing retention mechanisms and results in an improved peak shape for metabolites. The improved method using equimolar concentrations as well as the Z-HILIC column can potentially enable more reliable detection and improved peak shapes for more metabolites.

We uploaded the data from both experiments as a chromatogram library to Panorama Web to help identify metabolites in future studies.

### 3.2. Overview of ISAS Quant Metabolomics Library (IQML)

We utilized the existing Skyline small molecule framework for quantification [31], AutoQC for system suitability [38], and the Panorama web portal for sharing data publicly [30,38]. The workflow was optimized for MS1 level quantification from HRMS full scan mode data with RT and peak integration parameters curated manually from authentic metabolite standards data (Figure 1). Firstly, the transition lists, including precursor molecule name, mass (*m*/*z*), formula, retention time, and charge for both negative and positive ionization modes, are prepared and imported into the Skyline small molecule interface. The acquired raw MS data were imported to Skyline for peak extraction (EIC), integration, and retention time adjustment with a range of ±0.7. It was found that about 50% of metabolites had less than 0.5 ppm difference from the standard *m*/*z* values, and only up to 7% of metabolites have differences greater than 2 ppm. It showed that RT deviation for the standard metabolites is between 0.7% to 11% among all runs (*n* = 120) each for the positive and negative mode, respectively. Furthermore, it showed that Skyline provides a highly flexible environment for getting started with small molecules, even though it was first developed for peptides and targeted proteomics [30]. Our study also allows researchers to access the workflow and Skyline documents from the Panorama web portal to perform targeted metabolomics data analysis acquired on a similar analytical set-up.

Although there are several mass spectral libraries publicly [34,35,39] and commercially [40,41] available with tandem (MS/MS) data and RT data for metabolite and lipid standards, the quantitative information about these metabolite standards, such as linearity and response factors, are missing. This limits the application of current libraries only for metabolite identification, with no implications for large-scale metabolite quantification. We compiled the quantitative data from 100 metabolites to construct a quantitative library (IQML), which consists of a chromatogram, precursor *m*/*z*, and calibration curve for eight concentrations on the Panorama web portal, along with their RRF values reported separately. Figure 2 displays the workflow and coverage of the library metabolite KEGG pathways, which were assigned using Genome Pathway Mapper [42]. As shown, the study covered multiple important metabolite pathways. The library contains a total of 95 small molecules and their precursors/transitions with 140 replicates from 95 calibration curves. Additionally, there were six stable-isotope-labeled internal standards used for quantification as reference compounds (Table 1).

**Figure 2 metabolites-13-00844-f002:**
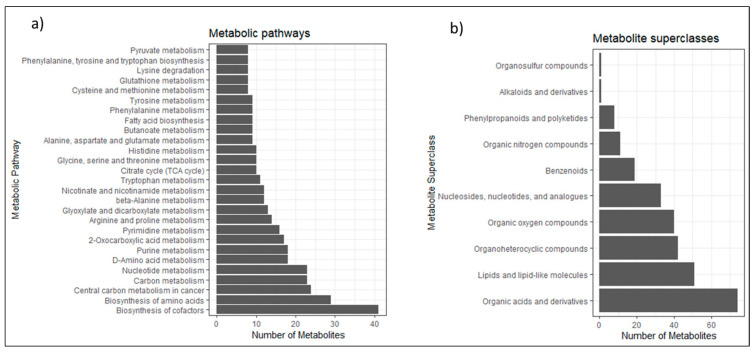
Overview of metabolite coverage of IQML: (**a**) Metabolic KEGG pathways covered in our metabolite library showing the number of metabolites from important metabolic pathways, which are included. The classification has been performed using KEGG pathway Mapper. (**b**) Metabolite superclasses covered in our metabolite library showing the number of metabolites from important metabolite superclasses, which are included. The classification has been performed using ClassyFire [43].

**Figure 3 metabolites-13-00844-f003:**
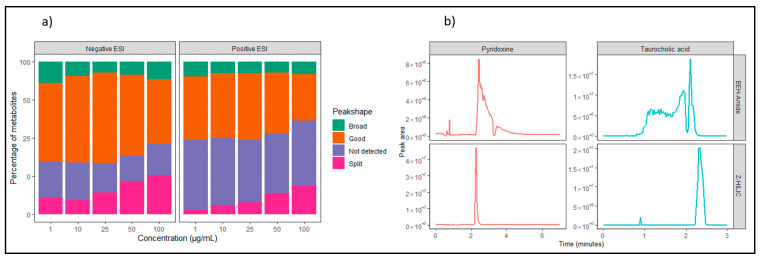
Comparison of BEH-Amide and Z-HILIC column: (**a**) Manual peak curation of 281 metabolites in positive and negative ESI using BEH-Amide column. The peak shape was manually classified as “Broad”, “Good”, “Not detected”, or “Split” at five concentrations. (**b**) Comparison of peak shapes in similar concentrations on BEH-Amide vs Z-HILIC column showing two metabolites with improved peak shape on Z-HILIC. Direct comparisons are shown between pyridoxine and taurocholic acid on BEH-Amide and Z-HILIC columns.

#### Quality Control, System Suitability, and Reproducibility

To monitor the system’s suitability before measurement, an automated quality control (AutoQC) system was set up. A standard mixture of amino acids is injected into the system prior to analysis. The data is automatically analyzed using Skyline and uploaded to the Panorama web portal. Here, a comparison between different qualifiers, such as retention time, mass accuracy, precursor area, and TIC area, is performed and visualized over three months of analysis time. This enables a quick and direct comparison of system runs. The AutoQC will also be used in future studies to continuously monitor the system performance and suitability prior to any experiment. In this study, no significant shift in these qualifiers could be observed, and the system was found to be suitable for the metabolomics analysis. The metabolites in plasma were identified using Progenesis QI. Twenty-eight compounds from our library were identified in negative ESI. The peak areas were analyzed to evaluate the precision of the data. Eighty-two percent of metabolites have a coefficient of variation (CV) of their peak areas below 15% across all replicates. Only four metabolites have a CV above 20%, indicating that the data is precise, and the measurements and instrument have been stable.

### 3.3. Quantitative Approach and Method Validation

#### 3.3.1. External Calibration Curve Evaluation for Linearity, Limit of Quantification (LOQ), and Dynamic Range

A total of 95 metabolites were detected in positive ESI and 93 in negative ESI. The linearity data revealed saturation effects of the generated calibration curves, which were more prominent in positive ESI. As shown in Figure 4a, we found that in negative ESI, nearly 80% of the metabolites had an R^2^ value greater than 0.95, and only four of the metabolites had R^2^ values below 0.90. In contrast, in positive ESI, 60% of the metabolites had R^2^ values greater than 0.95, and 25% of metabolites showed an R^2^ between 0.95 and 0.90, respectively, while 11 metabolites had R^2^ values below 0.90. While constructing calibration curves, the concentration levels that had bias < 15% (accuracy between 85% and 115%) were considered, and the points which were not in these criteria were excluded. For subsequent quantitative analysis, only metabolites with an R^2^ value above 0.9 were considered. To find out the instrumental variability, the CV values from different injections were calculated for each concentration. As shown in Figure 4b, most of the concentrations had an average CV < 30%. Furthermore, the mass error in ppm was calculated for our data. Figure 4c shows that in negative ESI, most metabolites showed a ppm error below 0.5. In positive ESI, the average ppm error lies between 0.5–1 ppm. Both ESI modes show a low ppm error, with only a minority of metabolites showing a ppm error above 2 ppm.

**Figure 4 metabolites-13-00844-f004:**
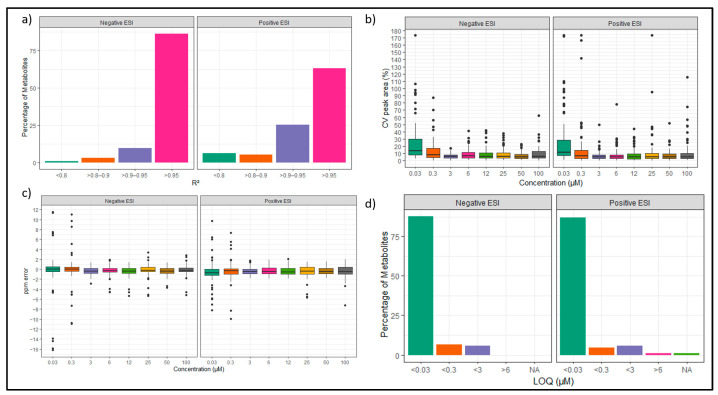
LC-MS performance parameters obtained from the calibration curves prepared at eight concentrations (*n* = 3): (**a**) Comparison R*^2^* values of the weighted regression of 100 metabolites in Z-HILIC in positive and negative ESI showing the number of metabolites in four different R*^2^* ranges. (**b**) Boxplot of the comparison of the CV peak area of standard measurements in triplicates using Z-HILIC column at eight different concentrations. (**c**) Comparison of calculated LOQ values of standards using Z-HILIC column in five categories in negative and positive ESI. (**d**) Boxplot of the comparison of the ppm error of standard measurements in triplicates using Z-HILIC shown at eight different concentrations in negative and positive ESI. The black dots in boxplot (**b**,**c**) show single data points which can be considered outliers.

The difference in the positive and negative ESI data can be explained by the method being more suited for negative ESI due to the basic pH acetate buffer system. Additionally, differences in metabolite ionization efficiencies can contribute to variations in the dynamic range and linearity in positive compared to negative ESI. The data revealed response saturation effects for the calibration curves, which were more prominent in positive ESI. This can be further investigated for their physicochemical properties and MS response. This could also be due to higher noise levels in positive data or enhanced ionization efficiency of protonated ions of those metabolites, as well as a non-suitable concentration range. To account for this observation in quantification, a weighted regression was performed using 1/x as a weighting factor. In negative ESI, LOD (limit of detection) above zero was calculated for 40.0%; in positive ESI, only 32.0% of metabolites have an assigned LOD above zero. As the LOD was identified using Skyline, it is defined as the concentration at which S/N > 2 and calculated through extrapolation of the line of regression. This can potentially lead to negative LOD values, which were not considered. Additionally, LOQ (limit of quantification) was defined for each metabolite as the lowest concentration at which it is identified. Figure 4d shows the LOQs calculated for the metabolites. The majority of metabolites with an assigned LOQ show an LOQ between 0.03 µM and 0.3 µM. Since these LOD and LOQ values are only mathematically calculated, they need to be confirmed experimentally in future studies.

In conclusion, we found a linear relationship for most of the metabolites, indicating the data are suitable for quantification. The lines of the equation of metabolites were now used for external calibration-based and RRF-based quantification of standard data and 28 identified metabolites from our library in plasma samples.

#### 3.3.2. Relative Response Factor Evaluation in Standards

The relative response factor approach considers a compound’s specific response in our LC-MS method relative to that of a non-endogenous reference compound. Hereby, the ion suppression and matrix effect of the compound of interest can be explored while considering a compound’s specific ionization efficiency.

As described in the Section 2, the internal standards selected as reference compounds (Table 1 and Appendix A) and their corresponding response factors (RF) were calculated for negative and positive ESI, as shown in Figure 5a. The chosen reference compounds cover a broad range of MS responses or ionization efficiencies and were suitable for quantification of metabolites based on their RF values. Further, RRFs for all metabolites using six reference compounds independently were defined. After defining the specific RRFs for each compound, the data are now subsequently used to quantify samples.

**Figure 5 metabolites-13-00844-f005:**
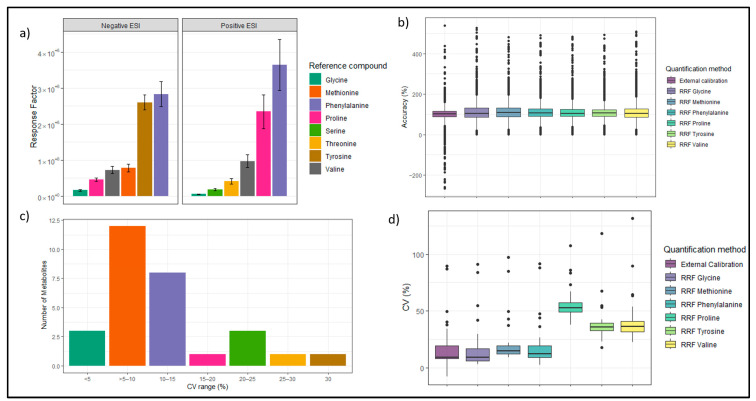
Response factor evaluation and comparison of quantification methods: (**a**) Response factors of defined reference compounds in standard measurements in negative and positive ESI showing six reference compounds in each ESI mode. (**b**) Boxplots showing accuracy of quantification methods (external calibration and RRFs) for the quantification (back calculation) of standard concentrations. (**c**) CV peak areas of 28 identified metabolites in plasma samples using progenesis QI in seven different categories. (**d**) Boxplots showing CV between triplicate quantifications of metabolites in plasma using RRF and external calibration. The black dots in boxplot (**b**,**d**) represent single data points which can be considered as outliers. (**e**) Plasma dependent ion suppression of reference compounds in plasma samples compared to pure solvents in negative and positive ESI showing six reference compounds in each ESI mode.

The RRF-based quantification approach was first tested using the standard calibration datasets. Here, the metabolite concentrations were back-calculated based on their peak area and their respective RRFs and additionally using external calibration quantification from Skyline data. Figure 5b shows boxplots of the accuracy values across the different quantification approaches (RRFs and external calibration curve) in negative ESI across all concentrations. The median of the external calibration approach as well as the RRF-based approach is close to 100%. For the external calibration, 60% of data points could be quantified with an accuracy of 80 to 120%. Using the RRF-based approach, between 49% and 57% of data points were quantified in the accuracy range. The smaller percentage can be explained with the uncertainty that is introduced when using (1) the slope as RF and (2) the average peak area of the reference compounds for the calculation of their RF. While using the external calibration, only the regression itself introduces uncertainty, as the calculated line equation does not represent all data points equally [6]. However, the data show that both quantification approaches are comparable for targeted metabolites. The RRF-based approach should have an advantage in terms of high throughput in the quantification of actual biological samples, which introduce matrix effects.

#### 3.3.3. RRF-Based Quantification of Plasma Compared to External Calibration

To quantify the metabolites in pooled human plasma samples, the targeted metabolites were identified using the Skyline workflow and a previous transition list of metabolites. Peak areas of 61 target metabolites in negative and 56 in positive ESI were extracted from all samples. Using Progenesis QI software (version v3.0 Waters Corporation, Milford, MA, USA) to re-analyze our data and evaluate the software performance of Skyline, we could identify 28 compounds in negative ion mode. We observed in Skyline, noise has been integrated as peaks, explaining the higher number of identified targeted metabolites. The quantification has been performed for the 28 compounds, which were identified in both Skyline and Progenesis QI. The concentrations of metabolites were quantified using the peak areas of the reference compounds and metabolites from the same run. The underlying assumption was that the reference compounds were introduced to the same instrumental conditions, matrix effects, and background noises as the plasma metabolites. The concentrations of metabolites in samples were calculated using all six reference compounds for positive and negative ESI. The averaged concentrations across all samples and replicates showed strong differences depending on which reference compound was used for quantification. Only five of those compounds showed a CV of the peak areas above 20%, as displayed in Figure 5c. The majority of identified metabolites showed a CV between 5% and 10%. Figure 5d shows the CVs of the quantified concentration among all replicates. Here, it shows that the RRF-based quantification using methionine or phenylalanine as reference compounds showed similar CVs as the external calibration-based quantification.

The quantified concentrations varied depending on the quantification approach or reference compound chosen. Generally, the RRF approach gives higher concentration values as it is used as a correction factor for the ion suppression effect in matrices and therefore corrects the quantification upwards. However, depending on the reference compound chosen, the concentration of the same metabolite varies greatly. To investigate this issue, the ion suppression of reference compounds in plasma samples compared to pure solvents was calculated using extracted data from Skyline, as shown in Figure 5e. Even though all reference compounds are amino acids, they showed ion suppression very differently. In positive ESI, this can be seen very clearly by comparing the ion suppression of phenylalanine (28.26%) with that of serine (77.93%). This again shows the complex nature of ion suppression effects in biological matrices and indicates that matching reference compounds to metabolites of interest purely based on their RF or chemical class may not be as reliable as the respective labeled internal standards. Our results showed that an RRF-based quantification is applicable to metabolomics studies as shown in the standard quantification data. However, matrix effects and the selection of reference compounds play a major role in the reliability of this approach. To investigate the reliability of our plasma quantifications, our results were compared to the literature.

#### 3.3.4. Benchmarking with Reference Plasma Material (NIST 1950 SRM)

To further validate our quantification approach, we compared our calculated metabolite concentrations with the NIST 1950 human plasma standard reference material (SRM) [44]. However, only literature concentrations for 12 metabolites overlapping with our library are available in the SRM. Although the Merck human plasma samples used and the NIST SRM are both pooled human plasma, they are from different sources and may not be comparable. However, the literature comparison results are shown in Appendix A. No clear pattern could be identified for the reference compound that gave the most reliable results. This was expected due to complex ion suppressions from different samples, even from the same biological sample type. Using a heavy-labeled form of a metabolite to quantify the natural form of the same metabolite gave realistic results in terms of the comparability of reported and calculated concentrations. In positive ESI, using the heavy-labeled version of a compound as a reference compound leads to accuracy values between 36% and 86% for the compound’s natural form compared to the NIST reference, while in negative ESI, the accuracy lies between 54% and 131%. This confirms the requirement of using matching labeled internal standards for reliable quantification.

It can be concluded that an RRF-based quantification approach may improve the reliability of metabolite quantifications in complex biological samples if a suitable reference compound is chosen. These compounds need to be selected carefully to best represent the compound of interest and its specific ionization and matrix suppression effects.

### 3.4. Limitations and Outlook of the Study

The RRF-based quantification approach has limitations regarding the selection of reference compounds. The results of this work showed that reference compounds have different ion suppression effects, in particular biological matrices, which are independent of their chemical class, mass, and even chromatographic retention times. It is not practical to match reference compounds to metabolites of interest purely based on a similar response in ESI or even the chemical class. All reference compounds used in this study were heavy-labeled amino acids, but the ion suppression of these compounds varied greatly. This means that the assigned reference compounds do not represent the ion suppression and matrix effects of the metabolite of interest, if not chosen correctly. The RRF should be calculated relative to a reference compound facing similar ion suppression mechanisms as the analyte to serve as a reliable correction factor. To address this problem, a future approach should contain multiple reference compounds and match them based on multiple qualifiers such as *m*/*z*, RT, metabolite class, RF, or functional groups. A rigorous quality control system with benchmarking for each sample type is required during the initial method development while implementing RRF-based quantification methods.

We also observed limitations of the Skyline software while performing automated data analysis for peak area integration and quantification. The validation parameters as calculated and defined by Skyline (LOD, LOQ) are defined loosely. The Skyline parameters are not universal for all types of peak shapes, and it also does not take into account RT shift due to sample types and concentration changes. To investigate this, we re-analyzed the plasma data in negative ion mode using Progenesis QI software for 28 compounds and two reference compounds. A striking difference in peak areas from each software was observed. After a manual evaluation of the integrated peaks in Skyline, we found that noise was integrated as peaks for multiple metabolites, explaining the difference in quantification and CVs obtained from both software.

The availability of such labeled internal standards for large-scale RRF-based quantification and validation of the method limits the implementation of the workflow across different labs. However, once established, our RRF approach still holds the potential to address metabolite quantification challenges and may facilitate translation into biomedical applications in the near to long term. The advantage of the approach, if evaluated further, is that it combines stable isotope quantification and external calibration methods. While EC does not consider the matrix effect, stable-isotope-labeled quantification usually does not consider the individuality of a compound’s response in LC-MS analysis. Using the RRF approach, both factors are considered. There are an increasing number of studies using the single point calibration approach (similar to RRF) due to these advantages. In addition, this approach is routinely used in lipidomic research [45,46,47] and pharmaceutical analysis [48,49,50]. Our approach will further facilitate the use of RRF in broader metabolomics research.

## 4. Conclusions

In this study, we present an analytical and computational workflow for large-scale quantification of targeted metabolites in human plasma samples. Initially, we developed a zwitterionic sulfobetaine HILIC method and compared its chromatographic performance with the widely used amide HILIC method for 281 metabolites. For metabolite quantification, we proposed the use of an RRF-based approach and benchmarked it against conventional EC and literature reports. Our findings revealed that, despite limitations, an RRF-based quantification approach is applicable and can be evaluated for human plasma metabolomics studies. The approach is potentially applicable to different sample types and matrices such as tissue or other biofluids. However, the reliability of this approach depends on matrix effects and the selection of suitable reference compounds. We conclude that the selection of reference compounds should be more considerate towards the type of biological matrix than the properties of the targeted metabolites. Therefore, an RRF-based quantification approach has the potential to enhance the reliability of metabolite quantification in complex biological samples if a suitable reference compound is chosen. Finally, we are sharing our results as a quantitative metabolomics library that other researchers can use to implement this approach for the quantification of over 250 metabolites in other biological sample types. In the future, RRF-based quantification can become more common for targeted metabolomics due to its advantages in rapid quantification and accuracy, a practice similar to lipidomics and pharmaceutical research.

## Figures and Tables

**Figure 1 metabolites-13-00844-f001:**
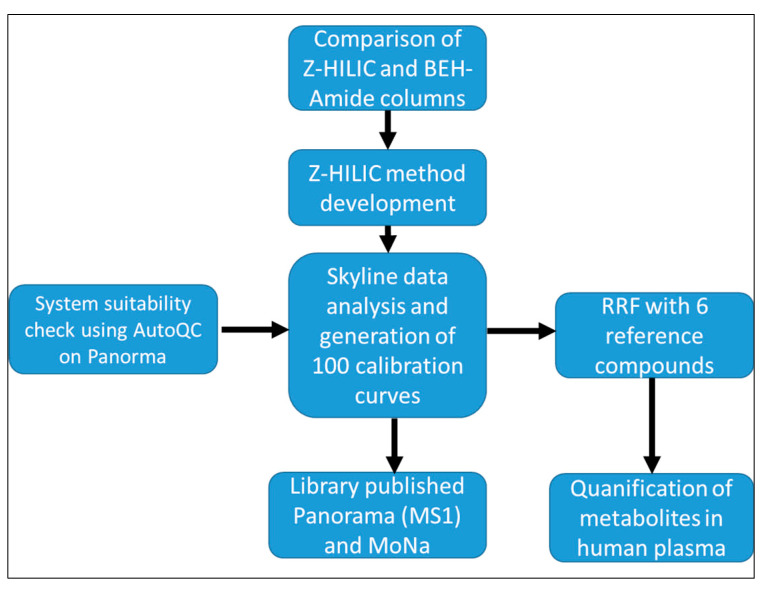
Schematic representation of study design: First, a comparison of two HILIC columns is performed, and the LC-MS method is developed with the z-HILIC column. The method is used to generate calibration curves for approximately *n* = 100 authentic metabolites standard using Skyline. The data is published on Panorama Web and MoNa as ISAS Quant metabolomics library (IQML) and was additionally used to quantify 28 metabolites in human plasma using the relative response factor (RRF) approach and 6 different reference compounds. Prior to any experiment, an automated system suitability control (AutoQC) using the Panorama Web application is introduced to the system.

**Table 1 metabolites-13-00844-t001:** List of standards used in this study, showing the chemical metabolite class, the use of the compounds in this study, and the number of metabolites assigned to each class.

Metabolite Super Class	Use	Number of Metabolites
Organic acids and derivatives	Standard	74
Lipids and lipid-like molecules	Standard	51
Organoheterocyclic compounds	Standard	42
Organic oxygen compounds	Standard	40
Nucleosides, nucleotides, and analogues	Standard	33
Benzenoids	Standard	19
Organic nitrogen compounds	Standard	11
Phenylpropanoids and polyketides	Standard	8
Alkaloids and derivatives	Standard	1
Organosulfur compounds	Standard	1
13C Amino acids	Internal standard/Reference compound	17

## Data Availability

Mentioned in the Section 2.

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
