# Peer review of "Quantitative Analytical and Computational Workflow for Large-Scale Targeted Plasma Metabolomics"

_metabolites, 2023, doi:10.3390/metabo13070844_

Round 1

Reviewer 1 Report

Widely quantification of metabolites is a major challenge for the metabolomics field especially for clinical and translational researchers. In this manuscript, the authors tried to establish a workflow for the quantification of 90+ metabolites using HILIC-based LC-MS coupled with the data analysis in Skyline software. The topic is very interesting to the audience. However, the results only showed the potential of quantitative metabolomics. This procedure showed high variations and is not applicable to another lab as a library. The detailed comments are as followings.

 1.For a good calibration curve, most of the time the R2 will be around 0.99 and at least higher than 0.95. The authors reported only about 60% of metabolites with an R2 above 0.9, which is not acceptable. This means that your calibration by itself yields a 10% variability error in your measurement (plus other steps such as sampling, sample treatment, and so on, the total variability will be much higher). The low R2 might be due to the concentration range. One of the challenges of wide metabolomics quantification is the concentration range of different metabolites in the samples. Since the aim of this study is to build a quantitative library, the variation will be much larger when other labs use this library.

 2. Usually, LOD should be defined to be the smallest amount or concentration of the analyte in the test sample that can be reliably distinguished from zero. S/N over 2 is a pretty loose definition. The authors run the samples of the concentration range between 0.03 to 100uM (line 221), how do they get the LOD of 0.01uM (line 391)? Please explain this. Also, what do the authors mean by LOD above zero (lines 385-386)?  Why do they get negative LOD?

3. The quantitative results should be compared with other traditional results such as amino acid analyzer or stable isotope dilution other than literature (results section, page 10).   

4. There are 3 metabolites (Indole-3-carboxylic acid, 3-Indoleacetonitrile, and 2,5-Furandicarboxylic acid in S-table 2) were identified to be eluted at a different time in the positive and negative modes. The authors should recheck the identification for them. If these were confirmed with authentic standards, what will cause the difference in retention time?

 5. Why authors selected 90+ metabolites for z-HILIC from those 260+ metabolites in the Amide column? What are the criteria? The authors reported 44% of peaks were classified as good in the positive and 60% were good in the negative in the Amide column. At least over 100 metabolites were good in the Amide column which is more than all the compounds selected in the z-HILIC column. Are those unselected compounds can not be detected in the z-HILIC column?

 6. Are there any compounds only detected in the z-HILIC column and not in the Amide column?

Author Response

Response to Review comments:

Quantitative analytical and computational workflow for large-scale targeted plasma metabolomics

Antonia Fecke, Nay Min Min Thaw Saw, Dipali Kale , Siva Swapna Kasarla , Albert Sickmann , Prasad Phapale *

Reviewer 1: Comments and Suggestions for Authors

Widely quantification of metabolites is a major challenge for the metabolomics field especially for clinical and translational researchers. In this manuscript, the authors tried to establish a workflow for the quantification of 90+ metabolites using HILIC-based LC-MS coupled with the data analysis in Skyline software. The topic is very interesting to the audience. However, the results only showed the potential of quantitative metabolomics. This procedure showed high variations and is not applicable to another lab as a library. The detailed comments are as followings.

  • We thank reviewer for recognition of importance of our work. We also agree with high variation of RRF quantification and its limitations in transferring to other labs. We have addressed this issue in “Limitations of the study” and also elaborated it further in conclusion part. Although our method has high variation, creating such quantitative libraries and sharing the results will eventually encourage other labs to adopt this workflow and better address the issue through inter-lab ring trials.

  1. For a good calibration curve, most of the time the R2 will be around 0.99 and at least higher than 0.95. The authors reported only about 60% of metabolites with an R2 above 0.9, which is not acceptable. This means that your calibration by itself yields a 10% variability error in your measurement (plus other steps such as sampling, sample treatment, and so on, the total variability will be much higher). The low R2 might be due to the concentration range. One of the challenges of wide metabolomics quantification is the concentration range of different metabolites in the samples. Since the aim of this study is to build a quantitative library, the variation will be much larger when other labs use this library.

  • We agree with the low R² values of our method. We have addressed potential causes of this issue and addressed the use of positive and negative ion mode. We have added further explanations to section 3.3.1 “External Calibration Curve Evaluation for Linearity, Limit of Quantification (LOQ), and Dynamic range”. Even though our method shows high variation we want to provide our data and encourage other scientists to work with this approach: “The difference in the positive and negative ESI data can be explained by the method being more suited for negative ESI due to the basic pH acetate buffer system. Additionally, differences in metabolite ionization efficiencies can contribute to variations in dynamic range and linearity in positive compared to negative ESI. The data revealed response saturation effects for the calibration curves, which were more prominent in positive ESI, as illustrated in Supplementary Figures S2. This can be further investigated for their physicochemical properties and MS response. This could also be due to higher noise levels in positive data or enhanced ionization efficiency of protonated ions of those metabolites, as well as a non-suitable concentration range.”

  1. Usually, LOD should be defined to be the smallest amount or concentration of the analyte in the test sample that can be reliably distinguished from zero. S/N over 2 is a pretty loose definition. The authors run the samples of the concentration range between 0.03 to 100uM (line 221), how do they get the LOD of 0.01uM (line 391)? Please explain this. Also, what do the authors mean by LOD above zero (lines 385-386)? Why do they get negative LOD?

  • We agree that the definition of LOD (S/N>2) as used in our Skyline method is loose and have addressed this issue in “Limitations and outlook” section: “We also observed limitations of the Skyline software while performing automated data analysis for peak area integration and quantification. The validation parameters as calculated and defined by Skyline (LOD, LOQ) are defined loosely. The Skyline parameters are not universal for all types of peak shapes, and it also does not take into account RT shift due to sample types and concentration changes.”

In this study, we utilized and improved on the Skyline small molecule workflow and quantifications. The LOD was calculated using this workflow. However, Skyline has limitations in statistical analysis for such calculation. In Skyline the LOD is calculated at the concentration where S/N>2 through extrapolation of the line of regression. This can cause negative LOD values. As negative LOD values are not to be considered, the LOQ has only been calculated for compounds with an LOD above zero.

  1. The quantitative results should be compared with other traditional results such as amino acid analyzer or stable isotope dilution other than literature (results section, page 10).
    • Yes, we agree with such orthogonal validation of results using traditional methods. However, we do not have access to such methods in-house. Also, in previous work (Ref 12) we found traditional methods (including amino acid analyzer) have low sensitivity compared to LC-MS analysis which makes comparison limited. We have compared the results for metabolite concentration values reported in literature for NIST reference material. The results are explained in “Benchmarking” section.
  1. There are 3 metabolites (Indole-3-carboxylic acid, 3-Indoleacetonitrile, and 2,5-Furandicarboxylic acid in S-table 2) were identified to be eluted at a different time in the positive and negative modes. The authors should recheck the identification for them. If these were confirmed with authentic standards, what will cause the difference in retention time?
  • We thank the reviewer for pointing out this mistake and we have corrected it.
  1. Why authors selected 90+ metabolites for z-HILIC from those 260+ metabolites in the Amide column? What are the criteria? The authors reported 44% of peaks were classified as good in the positive and 60% were good in the negative in the Amide column. At least over 100 metabolites were good in the Amide column which is more than all the compounds selected in the z-HILIC column. Are those unselected compounds can not be detected in the z-HILIC column?
  • Yes, they can be detected on Z-HILIC. However, we selected our metabolites based on the 3 criteria described in section 3.1. Metabolites which were not selected can potentially also be detected on Z-HILIC. One of the chosen criteria was “good/ broad peak shape in at least three concentrations” which is why the number of chosen metabolites is 100 even though at different concentrations a higher number of peaks were classified as good. We have changed Figure 3 to make this clearer and addressed this in section 3.1 “LC-MS Method Development”:

To address this issue (Figure 3 and Supplementary Figure S1), we repeated the measurement using equimolar concentrations of previously selected metabolites (Supplementary Table S2). To ensure reliable quantification and peak integration, metabolites were selected based on three criteria: 1) it is detected in at least three concentrations classified as "good" or "broad" (Supplementary Figure 1), 2) showing a linear relationship between concentration and peak area with an R² > 0.9 in at least three data points, and 3) not eluting in void volume with RT > 1 min. In total, we selected 100 metabolites using these criteria and measured them at eight concentrations ranging from 0.03 to 100 µM, using the BEH Z-HILIC column. We chose this column due to its expected ability to provide sharper peak shapes while providing a similar retention mechanism”

  1. Are there any compounds only detected in the z-HILIC column and not in the Amide column?
  • We agree that this would be an interesting comparison in terms of metabolite coverage, however in our study there are no compounds identified in Z-HILIC which were not previously found using the BEH-Amide column due to our experimental design. For our measurements on Z-HILIC column we only chose metabolites which were found, and their peak shape was classified as good in at least three or more concentrations on BEH-Amide column. We have changed Figure 3 as well as a description in section 3.1 “LC-MS Method Development” to make this clearer as also partially mentioned the reasons in response to the last comment. Text added: The improved method using equimolar concentrations as well as the Z-HILIC column can potentially enable more reliable detection and improved peak shapes for more metabolites.”

Reviewer 2 Report

Manuscript ID: metabolites-2410225
Type of manuscript: Article
Title: Quantitative analytical and computational workflow for large-scale targeted metabolomics

In this article, it proposes a new workflow for large-scale quantification of metabolites in human plasma samples, by adapting the Skyline software environment for processing such metabolomics data, and the results shared as a "quantitative chromatogram library" with the Panorama web application.

The authors report a novel quantitative chromatogram library with a targeted data analysis workflow for biomedical metabolomic applications, in order to facilitate large scale metabolite quantification, with an integrated analytical and computational workflow. In the other hand, and although this work have great potential in many bio-applications using a predetermined response factors or relative response factors (RRFs) mostly where authentic matrices and standards are difficult to obtain, the authors also highlighted its limitations. 

Therefore, under my point of view, this article provides an innovative solutions to analytical limitations and problems that are very important having in mind, the final clinical application of this methodology. However, further research would be necessary in order to broaden the field of application and for this new workflow to be used as a routine analytical tool.

I consider that, this article can be accepted for publication in Metabolites journal, taking into account the following comments, regard on:

·      2.1. Materials, Chemical, Reagent and Standards. 

This section should be tidied up and some information that is not relevant should be removed, such as Cat. No. Other information, for example 2.3.1. Preparation of standards should be appear in this section. 

·      2.2 Data availability and 2.4 Data Analysis 

Both sections could be together or maybe consecutive.

·      2.3 Methods. 

Only methods or procedures should be included.

·      2.4.3 Method validation parameters

Does it need to be in section 2.3 or can it be a different section?

·      3. Results and Discussion:

Section 3.1 LC-MS Method Development. The author used HILLIC metabolomics method  but this has been not discussed enough. 

Why method (reference 29: Nat. Protoc. 2012, 7, 872–881) is used in this work?

·      No table has been included thought the full text. 

The authors should consider including some tables that provide interesting data and not only include them in the supplementary material. Maybe Table 2 or 3 …?

Since I am not an expert in the development of an integrated analytical and computational workflow for quantitation using a predetermined response factors or relative response factors (RRFs), on the sections 3.2 and 3.3 I cannot contribute a lot. 

So I just make some minor suggestion, which the authors must be change through the full text: 

-   Units followed by data must be separated by a space:  for examples page 4 line 187 ïƒ  100 % water + 0.1 % ammonium acetate and page 5 lines 200 and 201 ïƒ  The ion transfer tube was heated to 300 °C, the 200 vaporizer temperature was set to 310 °C.

-   Page 5 line 217 ïƒ  The R2 is wrong; Regression coefficient is R

As I mentioned at the beginning, according to my criteria, the work presented is very interesting and has a lot of potential since it provides solutions to problems, effects and limitations that currently exist in metabolomics studies and that jeopardize the obtaining of quality data, in this specific case for the quantification of metabolites that could have biomedical applications of great interest.

Author Response

Response to Review comments:

Quantitative analytical and computational workflow for large-scale targeted plasma metabolomics

Antonia Fecke, Nay Min Min Thaw Saw , Dipali Kale , Siva Swapna Kasarla , Albert Sickmann , Prasad Phapale *

Reviewer 2: Comments and Suggestions for Authors

Manuscript ID: metabolites-2410225

Title: Quantitative analytical and computational workflow for large-scale targeted metabolomics

In this article, it proposes a new workflow for large-scale quantification of metabolites in human plasma samples, by adapting the Skyline software environment for processing such metabolomics data, and the results shared as a "quantitative chromatogram library" with the Panorama web application.

The authors report a novel quantitative chromatogram library with a targeted data analysis workflow for biomedical metabolomic applications, in order to facilitate large scale metabolite quantification, with an integrated analytical and computational workflow. In the other hand, and although this work have great potential in many bio-applications using a predetermined response factors or relative response factors (RRFs) mostly where authentic matrices and standards are difficult to obtain, the authors also highlighted its limitations.

Therefore, under my point of view, this article provides an innovative solutions to analytical limitations and problems that are very important having in mind, the final clinical application of this methodology. However, further research would be necessary in order to broaden the field of application and for this new workflow to be used as a routine analytical tool.

I consider that, this article can be accepted for publication in Metabolites journal, taking into account the following comments, regard on:

  • We Thank Reviewer for positive comments and would be happy to revise manuscript accordingly.
  • 2.1. Materials, Chemical, Reagent and Standards.

This section should be tidied up and some information that is not relevant should be removed, such as Cat. No. Other information, for example 2.3.1. Preparation of standards should be appear in this section.

  • Yes, we updated PCodes and made it consistent as per editor comments.

      2.2 Data availability and 2.4 Data Analysis

Both sections could be together or maybe consecutive.

  • Yes, now data availability section was moved to 2.4.5. under Data analysis.

  • 2.3 Methods.

Only methods or procedures should be included.

  • The section has been tidied up and includes only Methods now.

  • 2.4.3 Method validation parameters

Does it need to be in section 2.3 or can it be a different section?

  • We changed the title to “Method validation parameter calculations” to better reflect it place under Methods section. Since 2.4.3. only shows calculation of method validation parameters, so we think it is under appropriate section.

  • 3. Results and Discussion:

Section 3.1 LC-MS Method Development. The author used HILLIC metabolomics method but this has been not discussed enough.

Why method (reference 29: Nat. Protoc. 2012, 7, 872–881) is used in this work?

  • We agree that our method development section needed further details. For targeted metabolomics the HILIC protocol is the most used for polar metabolites, which we use routinely in our lab. However, we developed it further for better chromatography using z-HILIC as described in results. We address the advantages of HILIC in section 3.1 “LC-MS Method Development” as: “First, we optimized the widely used HILIC metabolomics method with a basic mobile phase (pH) and an XBridge Premier BEH-Amide column[37]. The HILIC method was chosen to ensure adequate retention for polar, hydrophilic or ionic compounds which cannot be properly separated using a reversed phase (RP) LC method, as used in multiple metabolomics studies. HILIC methods combine a stationary phase similar to that of a normal phase (NP) LC to help the retention of polar compounds while maintaining a mo-bile phase more similar to reversed phase LC methods. This helps analyzing compounds which cannot be separated using RP-LC but are not soluble in NP-LC mobile phases.”
  • No table has been included thought the full text.

The authors should consider including some tables that provide interesting data and not only include them in the supplementary material. Maybe Table 2 or 3 …?

  • Yes, we have included Table 1 which summarizes the covered classes of metabolites, reference compounds and their use in our study.

Since I am not an expert in the development of an integrated analytical and computational workflow for quantitation using a predetermined response factors or relative response factors (RRFs), on the sections 3.2 and 3.3 I cannot contribute a lot. So I just make some minor suggestion, which the authors must be change through the full text:

-   Units followed by data must be separated by a space:  for examples page 4 line 187  100 % water + 0.1 % ammonium acetate and page 5 lines 200 and 201  The ion transfer tube was heated to 300 °C, the 200 vaporizer temperature was set to 310 °C.

  • We thank the reviewer for pointing this mistake. We have changed this accordingly.

-   Page 5 line 217  The R2 is wrong; Regression coefficient is R2

  • We changed line to make this clearer and more correct as with reference: “The calibration curves were calculated with linear regression with weighting 1/(x*x) function from Skyline. The R² (coefficient of regression), CV (coefficient of variation) distribution and quantification of metabolites were represented as histogram or box plot using ggplot in R programming”

As I mentioned at the beginning, according to my criteria, the work presented is very interesting and has a lot of potential since it provides solutions to problems, effects and limitations that currently exist in metabolomics studies and that jeopardize the obtaining of quality data, in this specific case for the quantification of metabolites that could have biomedical applications of great interest.

  • We Thank again Reviewer for positive and encouraging report.

Reviewer 3 Report

The authors have done a consistent and systematic work in validating a targeted metabolomics assay and the use of relative response factors. This job can be important to the community as it evaluates the impact of matrix effects and ion suppression in several important metabolites.

As a reader i was interested in the presentation of the methods and the results but the manuscript would be more complete if this effect could be exlored in several types of samples like serum, types of plasma (edta, heparin, citrate), body fluids or tissues.

In any case, the methods are well presented, the analyses are solid but the results' presentation should be improved. The figures are not of great quality and not very self-explanatory.

Towards the end of the study, it feels like there is no enthousiasm to present the results in a way that the readers can come with general groups of metabolites and their behaviour or more particular ones.

Some comments on the text:

- line 19: RFF should be explained before use the acronym

- line 36: Claims about untargeted metabolomics can be challenged. add more references

- line 41: Targeted quantification is not always absolute

- line 47: Show some References

- line 167: Rephrase and be more clear

- The figures should be improved and the legends need to be much more explanatory

- line 272: This is a solid explanation. Did this improvement helped imeasuring more than 100  metabolites?

The English were fine.

Author Response

Response to Review comments:

Quantitative analytical and computational workflow for large-scale targeted plasma metabolomics

Antonia Fecke, Nay Min Min Thaw Saw , Dipali Kale , Siva Swapna Kasarla , Albert Sickmann , Prasad Phapale *

Reviewer 3: Comments and Suggestions for Authors

The authors have done a consistent and systematic work in validating a targeted metabolomics assay and the use of relative response factors. This job can be important to the community as it evaluates the impact of matrix effects and ion suppression in several important metabolites.

  • We Thank Reviewer for the appreciation of our work.

As a reader i was interested in the presentation of the methods and the results but the manuscript would be more complete if this effect could be exlored in several types of samples like serum, types of plasma (edta, heparin, citrate), body fluids or tissues.

  • We agree, that this would be an interesting point to address in this manuscript. We have applied the method in other biological samples and there is no limitation for its use in tissue or other body fluids collected in different forms. However, application of RRF in each matrix is lengthy work and may not fit into part and scope of this manuscript. We explained this under “Conclusion” section as: “Our findings revealed that, despite limitations, an RRF-based quantification approach is applicable and can be evaluated into human plasma metabolomics studies. The approach is potentially applicable to different sample types and matrices such as tissue or other biofluids. However, the reliability of this approach depends on matrix effects and the selection of suitable reference compounds.”

In any case, the methods are well presented, the analyses are solid but the results' presentation should be improved. The figures are not of great quality and not very self-explanatory.

  • We agree with this point. We have improved ALL Figures to make them more self-explanatory. Figure 4 (now Figure 5) has been redrawn and changed to a boxplot instead of a barplot. Additionally, we have redrawn all other figures to make their message more clear.

Towards the end of the study, it feels like there is no enthusiasm to present the results in a way that the readers can come with general groups of metabolites and their behaviour or more particular ones.

  • We redrawn ALL figures with legends for better clarity, added Table 1 to summarize the general groups of metabolites, reference compounds and added text about outlook and future directions under “Limitations and outlook of the Study” and Conclusion sections. We hope that these major modifications in figures, table and text are convincing enough to show our enthusiasm in explaining results and providing general guidelines for our readers.

Some comments on the text:

- line 19: RFF should be explained before use the acronym

  • We agree with this point and have modified the introduction of our acronyms. The acronym RRF is now first introduced in our abstract after a short introduction about the method: To facilitate large-scale metabolite quantification, we evaluated the relative response factor (RRF) approach combined with an integrated analytical and computational workflow. This approach considers a compounds individual response in LC-MS analysis relative to that of a non-endogenous reference compound to correct matrix effects.”

The acronym RF is now first introduced and explained in section 2.4.4 Quantification using Relative response factors (RRFs): Microsoft Excel was used on Skyline data output (peak areas and annotations) to define the response factor (RF). The response factor describes a compounds specific response in LC-MS analysis. Different compounds can exhibit a different level of response at the same concentration due to their ionization efficiency. This is described by the RF.“

- line 36: Claims about untargeted metabolomics can be challenged. add more references

à Yes, added relevant reference 5-7.

- line 41: Targeted quantification is not always absolute

  • We rephrased this sentence for more accuracy or clarity with new references: On the other hand, targeted metabolomics methods and data acquisition approaches can provide the absolute or relative quantification and validation of a chosen subset of metab-olites, based on a hypothesis-driven approach[8]. Targeted metabolomics allows re-searchers to circumvent the shortcomings of the global approach and provides more ac-curate and precise results which can complement with untargeted datasets[9].

- line 47: Show some References

à The references “8,9 an d10” were added”

- line 167: Rephrase and be more clear

  • We rephrased our sentence to make it clearer. The modified text is: For the preparation of pooled plasma samples commercially available lyophilized powder was reconstituted in water as per vendor instructions.

- The figures should be improved and the legends need to be much more explanatory

  • Yes, we have improved all our Figures to make them more self-explanatory with more detailed legends. Figure 4 (now Figure 5) has been redrawn and changed to a boxplot instead of a barplot. Additionally, we have redrawn our other figures to make their message clearer. We have also rephrased our legends and provided more details.

- line 272: This is a solid explanation. Did this improvement helped measuring more than 100 metabolites?

  • This improvement can help measure more metabolites in future. As we have shown in our data multiple more peaks can be detected using the HILIC method. Re-measuring these in a more suitable concentration range can help improve their peak shape and identification. We address this in line modified text is: “The improved method using equimolar concentrations as well as the Z-HILIC column can potentially enable more reliable detection and improved peak shapes for more metabolites.”

However, we did only measure metabolites on Z-HILIC which fell under this criteria in our BEH-Amide measurements: To address this issue (Figure 3 and Supplementary Figure 1), we repeated the measurement using equimolar concentrations of previously selected metabolites (Supplementary Table 2). To ensure reliable quantification and peak integration, metabolites were selected based on three criteria: 1) it is detected in at least three concentrations classified as "good" or "broad" (Supplementary Figure 1), 2) showing a linear relationship between concentration and peak area with an R² > 0.9 in at least three data points, and 3) not eluting in void volume with RT > 1 min. In total, we selected 100 metabolites using these criteria and measured them at eight concentrations ranging from 0.03 to 100 µM, using the BEH Z-HILIC column. We chose this column due to its expected ability to provide sharper peak shapes while providing a similar retention mechanism”

This has also been addressed with modifed section under “Limitations and Outlook of the Study”.  We hope this will address the reviewers concern appropriately.

Round 2

Reviewer 1 Report

The concept is ok for me. It is good to have a library with quantification indications. However, the results did not support the assumption well. Too much variation in the assay makes the proposed procedure not reliable. If the R2 for the standard curves is around or less than 0.9 for most of the compounds, there might be something wrong with the instrumental analysis or the data processing. Based on my own experience, the R2 reaching 0.95 and higher for the standard curve is not difficult on the LC-MS analysis. The authors need to investigate their method and the parameters of Skyline. The authors could confirm the integration of the linear curve with other software. If everything is correct, then the concept might not be suitable for untargeted metabolomics quantification due to the variation. 

Author Response

Response to reviewer's comments:
The concept is ok for me. It is good to have a library with quantification indications. However, the results did not support the assumption well.

--> This is a very general remark and does not align with 1st review report. We think this comment is specific to R2 values, which we addressed below.

Too much variation in the assay makes the proposed procedure not reliable.

--> We do not understand these comments as there is no "too much" variation in terms of high CVs and precision anywhere in the results. We think this comment is specific to R2 values which we addressed below.

If the R2 for the standard curves is around or less than 0.9 for most of the compounds, there might be something wrong with the instrumental analysis or the data processing. Based on my own experience, the R2 reaching 0.95 and higher for the standard curve is not difficult on the LC-MS analysis. The authors need to investigate their method and the parameters of Skyline. The authors could confirm the integration of the linear curve with other software. If everything is correct, then the concept might not be suitable for untargeted metabolomics quantification due to the variation.

--> We thank the reviewer for the constructive comments and suggestions. We agree with the reviewer about this issue of low R2 values and we address this to the satisfaction (all R2 > 0.9) and changed results, figures, and text accordingly. After a thorough investigation and manual curation of linearity data, we found that this was the issue with the Skyline software parameter which is already discussed in previous revision (linearity equation and peak integration). Particularly, we have changed the weighting factor of our linearity curves to better represent our data points. Additionally, we have manually integrated data points which were not previously properly integrated by the software. For subsequent analysis we have removed all metabolites with R² below 0.9.

We have now changed the quantification with a new linearity equation, recalculated the quantification, and redrawn figures accordingly. Since all R2 are not above 0.9 and most of them are above 0.95, we think the reviewer's concern is now addressed.

The following text has been updated in the revised manuscript Results section 3.3.1

A total of 95 metabolites were detected in positive ESI and 93 in negative ESI. The linearity data revealed saturation effects of the generated calibration curves, which were more prominent in positive ESI. As shown in Figure 4a, we found that in negative ESI, nearly 80 % of the metabolites had an R² value greater than 0.95, and only 4 of the metabolites had R² values below 0.90. In contrast, in positive ESI, 60 % of the metabolites had R² values greater than 0.95 and 25 % of metabolites showed an R² between 0.95 and 0.90, respectively, while 11 metabolites had R² values below 0.90. While constructing calibration curves, the concentration levels that had bias < 15 % (accuracy between 85 % and 115 %) were considered, and the points which were not in these criteria were excluded. For subsequent quantitative analysis only metabolites with an R² value above 0.9 were considered. “

“The data revealed response saturation effects for the calibration curves, which were more prominent in positive ESI. This can be further investigated for their physicochemical properties and MS response. This could also be due to higher noise levels in positive data or enhanced ionization efficiency of protonated ions of those metabolites, as well as a non-suitable concentration range. To account for this observation in quantification, a weighted regression was performed using 1/x as a weighting factor. In negative ESI, LOD (Limit of Detection) above zero was calculated for 40.0 %, in positive ESI only 32.0 % of metabolites have an assigned LOD above zero. As the LOD was identified using Skyline, it is defined as the concentration at which S/N>2 and calculated through extrapolation of the line of regression. This can potentially lead to negative LOD values, which were not considered. Additionally, LOQ (Limit of Quantification) was defined for each metabolite as the lowest concentration at which it is identified. Figure 4d shows the LOQs calculated for the metabolites. The majority of metabolites with an assigned LOQ show an LOQ between 0.03 µM and 0.3 µM. Since these LOD and LOQ values are only mathematically calculated, they need to be confirmed experimentally in future studies.“

Reviewer 3 Report

I think the manuscript can be accepted

Author Response

Response to the reviewer's comments

I think the manuscript can be accepted

--> We thank the reviewer for the acceptance of the revised manuscript. 

Round 3

Reviewer 1 Report

I appreciate the authors taking the time to address my questions. The revision meets my expectation and can be accepted as it is.